# Adoptive Cell Therapy in Hepatocellular Carcinoma: A Review of Clinical Trials

**DOI:** 10.3390/cancers15061808

**Published:** 2023-03-16

**Authors:** Muhammet Ozer, Suleyman Yasin Goksu, Baran Akagunduz, Andrew George, Ilyas Sahin

**Affiliations:** 1Department of Medical Oncology, Dana Farber Cancer Institute, Harvard Medical School, Boston, MA 02215, USA; 2Division of Hematology and Oncology, Department of Medicine, University of Texas Southwestern Medical Center, Dallas, TX 75390, USA; 3Department of Medical Oncology, School of Medicine, Erzincan Binali Yildirim University, Erzincan 24100, Turkey; 4Laboratory of Translational Oncology and Experimental Cancer Therapeutics, The Warren Alpert Medical School, Brown University, Providence, RI 02915, USA; 5Department of Pathology and Laboratory Medicine, The Warren Alpert Medical School, Brown University, Providence, RI 02915, USA; 6Legorreta Cancer Center, The Warren Alpert Medical School, Brown University, Providence, RI 02915, USA; 7Division of Hematology and Oncology, Department of Medicine, University of Florida, Gainesville, FL 32608, USA; 8University of Florida Health Cancer Center, Gainesville, FL 32608, USA

**Keywords:** adoptive cell therapy, chimeric antigen receptor, immunotherapy, T cell receptor, hepatocellular carcinoma (HCC), clinical trials

## Abstract

**Simple Summary:**

Immunotherapy has become the standard frontline treatment for patients with advanced hepatocellular carcinoma (HCC). However, drug resistance is a major limitation and novel strategies are needed for better clinical outcomes. In this review, we discuss adoptive cell therapy (ACT) as an emerging cancer therapy in the treatment of HCC and provide a summary of ongoing clinical trials.

**Abstract:**

Hepatocellular carcinoma (HCC) is the most common type of primary liver cancer. Immune checkpoint inhibitors (ICIs) have become the new reference standard in first-line HCC treatment, replacing tyrosine kinase inhibitors (TKIs) such as sorafenib. Many clinical trials with different combinations are already in development to validate novel immunotherapies for the treatment of patients with HCC. Adoptive cell therapy (ACT), also known as cellular immunotherapy, with chimeric antigen receptors (CAR) or gene-modified T cells expressing novel T cell receptors (TCR) may represent a promising alternative approach to modify the immune system to recognize tumor cells with better clinical outcomes. In this review, we briefly discuss the overview of ACT as a promising treatment modality in HCC, along with recent updates of ongoing clinical trials.

## 1. Introduction

Hepatocellular carcinoma (HCC), an inflammation-driven primary malignant cancer, is the seventh most common malignancy and the second most frequent cancer-related death worldwide [1]. Common risk factors include fatty liver disease (both alcoholic and non-alcoholic), alcohol consumption, chronic Hepatitis B and Hepatitis C, and toxins causing chronic inflammation [2]. While endemic viral hepatitis is the leading cause in African and Asian countries, chronic alcohol consumption and obesity-related non-alcoholic steatohepatitis (NASH) are the primary etiologies in Western countries [3]. The incidence of HCC is rapidly increasing, with an expected rate of increase of 2.8% annually through to 2030 [4]. Current standard treatment for HCC, categorized based on the stage of HCC, ranges from curative intent surgery and locoregional treatments to palliative systemic therapies. Unfortunately, most HCC patients are diagnosed with advanced disease and have dismal outcomes [5]. Based on American Cancer Society reports, the 5-year survival rate for patients with advanced HCC is less than 5% (last accessed 20 December 2022). Thus, novel therapeutic approaches for advanced disease are urgently needed.

HCC is a chemo-resistant malignancy with minimal clinical benefits when treated with various cytotoxic chemotherapies, interferon-alpha, and hormone therapies [6]. The protein tyrosine kinase inhibitors (TKIs) were the next treatment options that gained popularity after sorafenib received approval for advanced HCC in 2008 [7]. However, sorafenib and the next-generation TKIs, including lenvatinib, regorafenib, and cabozantinib have demonstrated only modest improvements in prognosis and have caused various toxicities [8,9,10].

As the liver has a diverse and complex immune microenvironment, HCC is considered an immunogenic tumor [11]. Along with cytotoxic T cells, the liver has various CD4+ T cells, regulatory T cells (Tregs), myeloid-derived suppressor cells (MDSCs), dendritic cells, and NK cells, which regulate responses to immunomodulatory therapies [12,13,14]. Studies showed an association between low recurrence rates and low CD4:CD8 ratios [15]. Therefore, immunotherapy is a promising modality for advanced HCC that could obtain durable and non-toxic anti-cancer activity. Over the past decade, development in immunotherapy has changed the landscape of advanced HCC treatment [16]. In 2020, FDA approved atezolizumab (anti-PDL1) combined with bevacizumab (VEGF inhibitor) for the treatment of patients with unresectable or metastatic HCC based on IMbrave150 trial data, showing better overall survival (OS) and progression-free survival (PFS) compared with sorafenib, and this regimen became a new first-line treatment for advanced HCC [17]. More recently, tremelimumab (anti-CTLA4) plus durvalumab (anti-PD-L1) demonstrated a statistically significant and clinically meaningful improvement in OS compared with sorafenib [18] and received FDA approval in October 2022. However, immune evasion and dysregulated immunity due to anti-inflammatory cytokines are major resistance mechanisms limiting current immunotherapy options’ effectiveness [2]. Therefore, novel immunotherapeutic strategies and the identification of therapy resistance are urgently needed. Current research focuses on the treatment of advanced HCC, mainly consisting of immune checkpoint inhibitors (ICIs), adoptive cell therapies, and tumor vaccines.

To date, with advancements in biotechnology, adoptive cellular therapy (ACT) is gaining attention as a novel treatment option in HCC. After the success in hematology, multiple clinical trials of chimeric antigen receptor T cells (CAR-T) and T-cell-receptor-modified T cells (TCR-T) therapies were conducted in patients with HCC. Studies suggest redirected cytotoxic cells can recognize HCC-tumor-associated antigens by inducing the programmed expression of synthetic T cell surface receptors [19]. Scientists aimed to increase the cytotoxic properties of the cells by engineering cytotoxic cells targeting HCC. The liver tumor microenvironment has an exceptional specificity to tumor-associated antigens, which makes HCC one of the most promising solid tumors for ACT [20]. In the context of TCR-T cell therapies, the recognition of MHC-restricted peptides is projected to be more adapted to solid-tumor management. Researchers hypothesized that replacing endogenous TCR with transgenic TCR will likely improve biological activity and increase clinical efficacy.

In this review, we discussed the overview of ACT in HCC along with the recent updates from ongoing clinical trials of ACT in HCC patients. HCC vaccination strategies are beyond the scope of this paper, and the reader is referred to the literature for further details [21].

## 2. Background of Adoptive Cell Therapy in HCC

The liver possesses a distinct immune system that makes it conducive to the development of immunogenic tumors, such as HCC [11]. This is attributed to its location, which facilitates pathogen detection via the gut and processing by various phagocytic cells, including Kupffer cells, as well as innate immune cells, such as NKT and iNKT cells. Additionally, the liver contains multiple subtypes of CD4+ T cells with immunomodulatory roles and cytotoxic CD8+ T cells. Despite these immune cells, they often experience T cell exhaustion and are unable to effectively control advanced HCC by themselves [22,23]. For an effective anti-tumor immune response, there should be a balance between tumor cells’ ability to present tumor-associated antigens (TAA) to antigen-presenting cells (APCs) and immunosuppressive activity from the tumor microenvironment. Naturally, the immune system tends to enhance self-tolerance, including malignant cells. Thus, the effective endogenous anti-tumor response depends on the increased infiltration of T cells, which promotes the expression of TAAs and tumor-specific antigens (TSAs) [24]. Previous studies with HCC patients undergoing hepatic resection reported that prominent lymphocyte infiltration is associated with better OS and decreased recurrence [15,25]. However, the relationship between activated lymphocyte infiltration and survival rates needs further investigation. The non-gene-modified ACT includes cytokine-induced killer (CIK) cells and tumor-infiltrating lymphocytes (TILs). CIKs are CD3 + CD56 + natural killer (NK)-like T cells with non-MHC-restricted cytotoxic and proliferative activity [26,27]. On the other hand, TILs are polyclonal tumor-targeting T cells expanded to use as autologous therapy.

Recently, gene engineering technology has allowed modifying immune cells with synthetic receptors, such as CAR-T and TCR-T, to enhance the visibility of TAA and TSA. These tumor antigens are the primary targets used in ACT for HCC and mainly fall into one of the three categories, including tumor-associated antigens (e.g., AFP, GPC-3), viral-derived cancer antigens (VHB, VHC), and cancer–testis antigens (e.g., NY-ESO-1, MAGE). A schematic overview of major steps within the CAR-T and TCR-T adoptive cell therapy process is summarized in Figure 1.

To date, many ongoing early phase clinical trials of ACT dedicated to HCC are under evaluation.

## 3. Chimeric Antigen Receptor T Cells (CAR-T)

Chimeric antigen receptors (CARs) are synthetic cell surface receptors composed of an extracellular single-chain variable fragment, intracellular domain with a hinge region, a transmembrane region, and several domains from costimulatory molecules [28,29]. The extracellular domain recognizes membrane antigens expressed at the target cells. The hinge region provides flexibility and binds the related ligand precipitate T cell activation through the TCR domain. The intracellular signaling domain is modifiable for functional improvement. CAR-T cell therapy consists of engineered autologous T cells to recognize and eliminate malignant cells [30,31]. ACT aims to generate tumor-specific CAR-T cells, which are not restricted by the major histocompatibility complex (MHC) and could overcome immune escape [32,33]. Currently, three autologous CAR-T cell therapies have FDA approval for relapsed or refractory large B-cell lymphoma, multiple myeloma (MM), acute lymphoblastic leukemia (ALL), and non-Hodgkin lymphoma (NHL) [34,35,36].

As CAR-T cell therapy greatly impacted hematological malignancies, many studies were conducted to target solid tumors by CAR-T. After promising results from preclinical models, CAR-T cells targeting various TAAs have been developed and become one of the most encouraging immunotherapy options for HCC. Several promising CAR-T cell tumor targets were identified, including Glypican-3 (GPC3), alpha-fetoprotein (AFP), NK group 2 member D ligand (NKG2DL), Mucin 1 glycoprotein 1 (MUC1), Epithelial cell adhesion molecule (EpCAM), Claudin18.2 (CLD18), CD147, CD133, HBV surface protein, and c-MET [37,38,39]. Currently, 24 phase I/II CAR-T cell clinical trials for HCC are ongoing. While some clinical trials have incorporated lymphodepletion prior to adoptive cell therapy using mostly a combination of fludarabine and cyclophosphamide, others have not mentioned any lymphodepleting chemotherapy in their protocols. Most trials are at the early phase and evaluate safety as a primary endpoint. Secondary endpoints include overall response, response duration, and T cell persistence in peripheral blood. The ongoing clinical trials for CAR-T cell therapies have been summarized in Table 1.

Of the limited completed studies available, a phase 2 study evaluated CD133-directed CAR T cells in adults with advanced HCC that was refractory to treatment to investigate their safety and efficacy. Of the 21 patients, 1 had a partial response, 14 had stable disease, and 6 progressed after T cell infusion. The median OS was 12 months, and the median PFS was 6.8 months, with the most common high-grade adverse event being hyperbilirubinemia. The results suggest promising anti-tumor activity and a manageable safety profile for CART-133 cell therapy in previously treated advanced HCC [38].

Currently, most CAR-T cell therapies target glypican-3 (GPC3) as a treatment option for HCC [40,41,42]. There are 11 ongoing phase I/II clinical trials targeting GPC3, and one targeting GPC3/TGFβ (NCT03198546). GPC3 is overexpressed in HCC but has very limited to no expression in normal tissues, making it a great target for CAR-T cell therapies [43]. Multiple pre-clinical studies suggest GPC3 targeting with ACT [44,45]. One study explored the efficacy of human CAR-T cells engineered to secrete IL-7 and CCL19 (7 × 19) in the treatment of solid tumors, including HCC. The 7 × 19 CAR-T cells demonstrated superior expansion, migration, and tumor suppression capabilities in vitro and in xenograft models of HCC. A complete tumor disappearance was observed 30 days after an intratumor injection of anti-GPC3-7 × 19 CAR-T treatment in a patient with advanced HCC enrolled in an ongoing phase 1 clinical trial (NCT03198546) [46].

In their two sequential phase 1 clinical trials (NCT02395250; NCT03146234), Shi et al. reported the safety and anti-tumor activity of GPC3 CAR-T cells in GPC3+ HCC patients [47]. Cytokine release syndrome (CRS) was observed in nine patients, with one case being fatal grade 5 CRS. Although the rapid expansion of CAR-T cells in vivo after infusion has been linked to cytokine release syndrome and correlates with anti-tumor responses in hematological malignancies, solid-tumor CAR-T cell trials have not reported outcomes with the significant release of proinflammatory cytokines prior to tumor regression, which suggests that the unsatisfactory response rates observed in patients with solid tumors are likely due to the insufficient expansion and persistence of CAR T cells [48].

As another CAR-T target, AFP is a well-described biomarker for HCC, expressed in 60–80% of patients and related to poor prognosis [49]. After pre-clinical studies reported AFP as a potential target for the ACT, a phase I clinical trial was conducted in HCC patients (NCT03349255); however, the trial was terminated. Currently, a new T cell construct for the same indication is pending. NKG2DL is overexpressed in HCC and mainly absent in normal cells, highlighting its potential as a good target for CAR-T cell therapy [50]. In November 2021, a phase I NKG2D-based CAR-T cells clinical trial started recruiting patients (NCT05131763). The EpCAM has oncogenic potential and is over-expressed in various carcinomas, including HCC, colon cancer, prostate cancer, pancreas cancer, and esophageal cancer [51,52]. A CARTEPC trial (NCT03013712) was conducted to evaluate CAR-T cell therapy targeting EpCAM-positive cancers; outcomes have not yet been reported. Another undergoing basket study is targeting CLD18 (NCT03302403). MUC1 is overexpressed in HCC and correlated with prognosis in patients with HCC [53]. Pre-clinical studies validated MUC1 as a promising target for CAR-T therapy [54,55]. A phase I\II basket trial of MUC1-CAR-T cell therapy (NCT02587689) is underway in patients with HCC, breast, lung, and pancreatic cancer.

## 4. T-Cell-Receptor-Transduced T Cells (TCR-T)

As part of the adoptive cell therapy strategies, multiple studies focused on TCR-engineered T cells specifically designed to recognize targeted tumor cells via TAA/TSA and MHC-restricted peptides. This concept gives TCR-T the advantage of targeting intracellular antigens expressed on HLA [56,57,58]. On the other hand, it limits the therapy to the most frequently shared HLA types [59]. To date, TCR -T cell therapies mainly focus on viral-associated peptides and AFP as the optimal targets. To date, 11 phase I/II TCR-T HCC clinical trials are ongoing. These are early phase studies mainly looking at safety data, tolerability, and partial or complete response. The ongoing clinical trials for TCR-T cell therapies are summarized in Table 2.

An early phase trial that published preliminary results examined the effectiveness and safety of HBV-specific TCR-expressing autologous T cells with a short life span in patients with advanced HBV-related HCC who were not suitable for liver transplantation. The study enrolled eight patients, of whom two experienced adverse events; however, the treatment was generally safe and well tolerated. One patient achieved a partial response lasting for 27.7 months, and most of the patients exhibited a reduction or stabilization of circulating HBsAg and HBV DNA levels after this therapy. The results of this early study support the application of this treatment strategy in advanced HBV-related HCC patients [60].

High-affinity TCR-T cells can be induced with HBV infection and potentially be a good target for engineered TCRs. Integrated HBV-DNA in HCC cells include short segments that encode epitopes capable of activating T cells. By utilizing the HBV transcriptomes from these cells, they could be used to engineer T cells for personalized immunotherapy; therefore, this approach has the potential to treat a broader population of patients with HBV-associated HCC [61]. This phenomenon was initially shown in patients with HCC recurrence after liver transplantation. Overall, integrating HBV-DNA into the genome of HCC tumor cells enables the production of HBV antigens that TCR-T cells can recognize. The main concern is liver damage considering HBV-infected non-malign liver cells could obtain lyses. Earlier this year, Tan et al. reported a favorable safety profile and long-term clinical benefit of short-lived mRNA HBV-TCR-T cell therapy in primary diffused non-operable HBV-HCC [62]. Additionally, another study published in 2019 suggested HBV transcriptomes from HBV-HCC patients could be used to engineer T cells for personalized immunotherapy. They showed a good safety profile without adverse events occurring. To date, there are five phase I and two phase I/II clinical trials targeting HBV TCR-T cells.

As presented on HLA, AFP is also a good target for TCR-T cell therapies. Initial data from the genetically engineered autologous SPEAR T cells (AFPc332 T cells) show promising response rates [63]. The complete results of the study have yet to be published (NCT03132792). An additional two phase I AFP TCR-T cell clinical trials have been conducted since 2019, with no reported results (NCT04368182, NCT03971747). Another trial is currently active, evaluating the melanoma-associated antigen 1 (MAGEA1) as a TCR-T target (NCT03441100).

## 5. Conclusions and Future Expectations

Since the discovery of immunotherapies in HCC treatment, the landscape of the treatment has changed remarkably. ACT has emerged as a novel therapeutic approach as part of cellular immunotherapy strategies. Compared with immune checkpoint inhibitors, the ACT is still in the early development stages for HCC treatment. ACT is a new and evolving modality for HCC; therefore, many clinical trials are in phase I/II settings and focused on adverse events, dose-limiting toxicities, maximum tolerated doses, safety, and tolerability. To date, various clinical trials for the ACT are at an early stage and face multiple limitations. First, the complex tumor microenvironment causes biological and physical barriers for ACT to reach its targets [64,65]. For ACT to be effective, its target should be highly expressed on the tumoral cell surface, with very limited to no expression on healthy tissues. Studies showed that normal cells have the ability to express some of the solid-tumor targets, which brings the risk of on-target and off-tumor toxicity. The major safety challenges of ACT in HCC, especially for targeted T cell therapies, include the heterogeneity of TSAs, and TAAs across the tumors [66,67]. Future studies should focus on searching for HCC-specific antigens that could be excellent targets for the ACT. Down-regulation of MHC molecules and the short viability of the ACT cells are also limiting factors. Toxicities from the off-tumor effects of CAR-T and TCR-T cells are another significant concern in managing HCC [68,69,70,71]. Most adoptive cell therapy clinical trials were conducted in Child–Pugh class A patients, so shared safety data might not be definitive for sicker patients. To overcome unexpected toxicities from ACT, further research should evaluate genetic engineering interventions to optimize ACT affinity to target cells. In CAR-T cell therapy, having multiple HCC antigen targets potentially enhances specificity and improves safety.

More efforts are needed to focus on the improvement of the efficacy and safety of CAR-T and TCR-T cell therapies before confirmation in a phase 3 setting. Ultimately, we believe that treatments that combine different immunotherapy modalities are anticipated to improve clinical outcomes in patients with HCC.

## Figures and Tables

**Figure 1 cancers-15-01808-f001:**
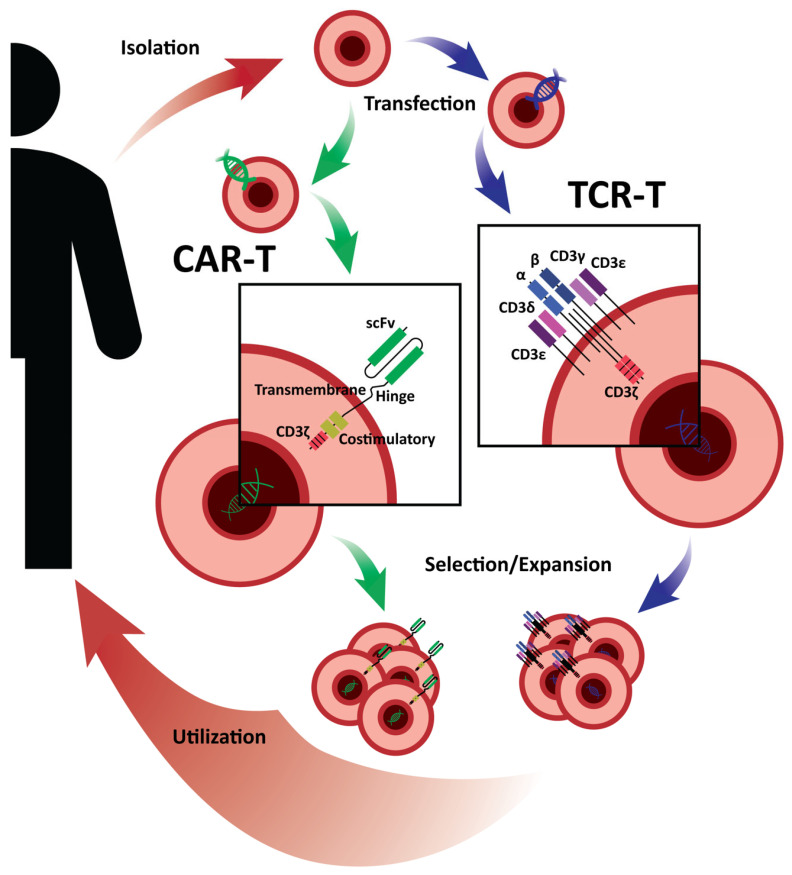
Schematic overview of major steps within CAR-T and TCR-T adoptive cell therapy process. T cells are isolated autologously from patient blood or from a healthy allogeneic source. Leukocyte separation is accomplished through leukocyte apheresis, allowing isolation of peripheral blood mononuclear cells (PBMCs). T cells are selectively stimulated prior to viral transduction of engineered CAR or TCR gene(s), or via alternate genetic editing tools. Successfully transfected clones are expanded for therapeutic use. Note: costimulatory domains on the CAR vary based on CAR generation.

**Table 1 cancers-15-01808-t001:** Ongoing clinical trials of chimeric antigen receptor (CAR) T cell therapies for hepatocellular carcinoma (HCC).

Agent	Status	Phase	Clinicaltrial.gov (accessed on 10 March 2023)	Sample Size	Patient Characteristics	Primary Outcome
GPC3 CAR-T	Active	Phase 1	NCT02905188	9	HCC	DLT, CR/PR
GPC3 CAR-T	Recruiting	Phase 1	NCT04121273	20	HCC	DLT
GPC3 CAR-T	Recruiting	Phase 1	NCT05070156	3	HCC	Safety and tolerability
GPC3 CAR-T	Recruiting	N/A	NCT05620706	20	HCC	AEs, ORR
GPC3 CAR-T	Unknown	Phase 1 and 2	NCT03130712	10	HCC	Safety and tolerability
GPC3 CAR-T	Recruiting	Phase 1	NCT05003895	38	HCC	Safety and feasibility
GPC3 CAR-T	Recruiting	Phase 1	NCT04951141	10	HCC, cholangiocarcinoma	AEs, ORR
GPC3 CAR-T	Unknown	Phase 1 and 2	NCT03084380	20	HCC	AEs, efficacy
GPC3 CAR-T	Recruiting	Phase 1	NCT05103631	27	HCC	DLT, CR/PR
GPC3 CAR-T	Not yet recruiting	Phase 1	NCT05344664	12	HCC	AEs
GPC3 CAR-T	Unknown	Phase 1 and 2	NCT02715362	30	HCC	Safety
GPC3 and/or TGFβ targeting CAR-T	Recruiting	Phase 1	NCT03198546	30	HCC SCLC	DLT CR/PR
EpCAM CAR-T	Recruiting	Phase 1	NCT05028933	48	HCC, CRC, pancreatic and gastric cancers	DLT, MTD, AEs
EpCAM CAR-T	Unknown	Phase 1 and 2	NCT03013712	60	HCC, colon, esophageal, prostate, pancreatic, and gastric cancers	Toxicity profile, survival time, and efficacy
CLD18 CAR-T	Active	N/A	NCT03302403	18	HCC, pancreatic cancer, esophageal cancer, MM, B-cell lymphoma, and B-cell leukemia	AEs, engraftment
MUC1 CAR-T	Unknown	Phase 1 and 2	NCT02587689	20	HCC, NSCLC, pancreatic carcinoma and triple-negative breast cancer	AEs
CD147 CAR-T	Unknown	Phase 1	NCT03993743	34	HCC	AEs, DLT, MTD
CAR-T/TCR-T	Recruiting	Phase 1 and 2	NCT03638206	73	HCC, renal, ovarian, esophagus, colorectal, lung, pancreatic, gastric cancers	AEs, clinical response
Bold is not necCAR-T/TCR-T	Recruiting	Phase 1 and 2	NCT03941626	50	HCC, glioma, gastric cancer, esophageal cancer	AEs, clinical response
NKG2D CAR-T	Recruiting	Phase 1	NCT05131763	3	HCC, glioma, medulloblastoma, colon cancer	AEs
NKG2D CAR-T	Not yet recruiting	Phase 1	NCT04550663	10	HCC, colorectal cancer, glioma	MTD, AEs
B7H3 CAR-T	Recruiting	Phase 1 and 2	NCT05323201	15	HCC	Safety, ORR
Armored CAR-T	Recruiting	Phase 1	NCT05155189	20	HCC	AEs
c-Met/PD-L1 CAR-T	Unknown	Phase 1	NCT03672305	50	HCC	Efficacy, AEs

DLTs: dose-limiting toxicities, CR: complete response, PR: partial response, AEs: adverse events, ORR: overall response rate, MTD: maximum tolerated dose.

**Table 2 cancers-15-01808-t002:** Ongoing clinical trials of T-cell-receptor-transduced T cell therapies for hepatocellular carcinoma (HCC).

Agent	Status	Phase	Clinicaltrial.gov (accessed on 10 March 2023)	Sample Size	Patient Characteristics	Outcome
AFP T cells	Unknown	Phase 1	NCT04368182	3	HCC	ORR, DCR, DOR
AFP T cells	Unknown	Phase 1	NCT03971747	9	HCC	AEs, ORR, DOR
AFP T cells	Active, not recruiting	Phase 1	NCT03132792	45	HCC	DLT, AEs, CR/PR
HBV-TCR T cells	Active, not recruiting	Phase 1	NCT04677088	7	HCC	AEs, ORR
HBV-TCR T cells	Recruiting	Phase 1 and 2	NCT05417932	47	HCC	AEs, tumor response
HBV-TCR T cells	Unknown	Phase 1	NCT02719782	10	HCC	Safety, efficacy
HBV-TCR T cells	Recruiting	Phase 1	NCT04745403	10	HCC	AEs, ORR
HBV-TCR T cells	Recruiting	Phase 1	NCT03899415	10	HCC	AEs, ORR
HBV-TCR T cells	Recruiting	Phase 1	NCT05339321	36	HCC	AEs, efficacy
HBV-TCR T cells	Not yet recruiting	Phase 1 and 2	NCT05195294	55	HCC	AEs, ORR
MAGEA1	Active, not recruiting	Phase 1	NCT03441100	15	HCC and multiple solid tumors	AEs, tumor response

ORR: overall response rate, DCR: disease control rate, DOR: duration of response, DLT: dose-limiting toxicities, AEs: adverse events.

## Data Availability

No new data were created.

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
