# Peer review of "Adoptive Cell Therapy in Hepatocellular Carcinoma: A Review of Clinical Trials"

_cancers, 2023, doi:10.3390/cancers15061808_

Round 1

Reviewer 1 Report (New Reviewer)

The authors of this manuscript summaries the different clinical trials utilising CAR and TCR-T cells for the treatment of HCC providing some indications about their efficacy. There is very little explanation about the rational of the different T cell products engineered to specifically target different tumor or viral antigens presented by HCC. On the other hand, this might not be the real focus of this review and as such this manuscript can constitute a  simple summary of what is clinically tested in this specific filed.

I have a couple of minor request:

a) the authors wrote:

"The immunogenicity of HCC makes it a good target for immunity-based therapies" How the authors can support such bold statement? Are we sure that HCC is highly immunogenic?  Some work analysing the "T cell immunogenicity of different classical tumor antigens in HCC found some T cell response but such T cells were mainly exhausted. ( see  

Gehring, A.J., Z.Z. Ho, A.T. Tan, M.O. Aung, K.H. Lee, K.C. Tan, S.G. Lim, and A. Bertoletti. 2009. Profile of tumor antigen-specific CD8 T cells in patients with hepatitis B virus-related hepatocellular carcinoma. Gastroenterology. 137:682–690. doi:10.1053/j.gastro.2009.04.045.) I also do not  think  that the presence of " activated lymphocytes can be used as a parameter of "immunogenicity". I think this section should be improved. 

2) For the use of HBV antigen as a tumor antigen , I think the authors should explain briefly why HBV can be used as a tumor antigen in HCC  (HBV-DNA can integrate in the genome of HCC cells and the integrated HBV-DNA can produce HBV antigens that can be recognised by TCR -T cells and also point out that  this strategy has been initially used in HCC relapses after liver transplantation and not only in primary HCC. ( see Tan, A.T., et al. 2019. Use of Expression Profiles of HBV-DNA Integrated Into Genomes of Hepatocellular Carcinoma Cells to Select T Cells for Immunotherapy. Gastroenterology. 156:1862-1876.e9. doi:10.1053/j.gastro.2019.01.251.)

Author Response

Response to Reviewers

We would like to take the time to thank the Editorial Board, Editorial Staff, and reviewer for their time and efforts in reviewing and processing our manuscript. We sincerely appreciate the efforts dedicated toward our manuscript and are grateful to have our work reviewed.  We have made every effort to address the reviewer's comments.

The Reviewer

The authors of this manuscript summaries the different clinical trials utilising CAR and TCR-T cells for the treatment of HCC providing some indications about their efficacy. There is very little explanation about the rational of the different T cell products engineered to specifically target different tumor or viral antigens presented by HCC. On the other hand, this might not be the real focus of this review and as such this manuscript can constitute a  simple summary of what is clinically tested in this specific filed.

I have a couple of minor request:

  1. a) the authors wrote:

"The immunogenicity of HCC makes it a good target for immunity-based therapies" How the authors can support such bold statement? Are we sure that HCC is highly immunogenic?  Some work analysing the "T cell immunogenicity of different classical tumor antigens in HCC found some T cell response but such T cells were mainly exhausted. ( see  Gehring, A.J., Z.Z. Ho, A.T. Tan, M.O. Aung, K.H. Lee, K.C. Tan, S.G. Lim, and A. Bertoletti. 2009. Profile of tumor antigen-specific CD8 T cells in patients with hepatitis B virus-related hepatocellular carcinoma. Gastroenterology. 137:682–690. doi:10.1053/j.gastro.2009.04.045.) I also do not  think  that the presence of " activated lymphocytes can be used as a parameter of "immunogenicity". I think this section should be improved.

Author response: Thank you for your insightful feedback and for giving us the opportunity to elaborate further on this topic. The liver has its own distinct immune system, which makes it particularly susceptible to the development of immunogenic tumors like HCC (Kubes et al. PMID: 29328785). This is due in part to its location, which allows for pathogen detection through the gut and processing by various phagocytic cells, such as Kupffer cells, as well as innate immune cells like NKT and iNKT cells. Moreover, the liver contains various subtypes of CD4+ T cells with immunomodulatory roles, as well as cytotoxic CD8+ T cells. However, despite the presence of these immune cells, they often become exhausted and are unable to effectively control advanced HCC on their own (Rochigneux et al. PMID: 31413924). Nonetheless, the existence of tumor-associated antigens (TAA) with reasonable specificity makes HCC one of the most promising tumors for ACT (Comoli et al. PMID: 31435646). We have thoroughly explained this concept in our manuscript, cited the pertinent article you brought to our attention, and updated the corresponding section.

Regarding the correlation between activated lymphocytes and HCC, Unitt et al. reported a negative correlation between tumor lymphocyte infiltration and HCC recurrence rate, as well as better survival rates (Unitt et al. PMID: 16580084, Wada et al. PMID: 9462638). However, we agree that limited data is available to establish a direct correlation between immunogenicity and activated lymphocyte infiltration. We have accordingly revised this section in the manuscript, which can be found at Lines: 107-122.

Thank you once again for your valuable feedback and support.

2) For the use of HBV antigen as a tumor antigen , I think the authors should explain briefly why HBV can be used as a tumor antigen in HCC  (HBV-DNA can integrate in the genome of HCC cells and the integrated HBV-DNA can produce HBV antigens that can be recognised by TCR -T cells and also point out that  this strategy has been initially used in HCC relapses after liver transplantation and not only in primary HCC. ( see Tan, A.T., et al. 2019. Use of Expression Profiles of HBV-DNA Integrated Into Genomes of Hepatocellular Carcinoma Cells to Select T Cells for Immunotherapy. Gastroenterology. 156:1862-1876.e9. doi:10.1053/j.gastro.2019.01.251.)

Author Response: We would be delighted to provide further details on this topic in our manuscript, and we have cited the relevant article as suggested. It has been demonstrated that integrated HBV-DNA in HCC cells contains short segments that encode epitopes capable of activating T cells. By utilizing the HBV transcriptomes from these cells, T cells can be engineered for personalized immunotherapy, thus potentially treating a broader range of patients with HBV-associated HCC. This phenomenon was first observed in patients with HCC recurrence after liver transplantation. In general, integrating HBV-DNA into the genome of HCC tumor cells enables the production of HBV antigens that TCR-T cells can recognize. Please refer to the updated section of our manuscript, located between lines 239-246.

This manuscript is a resubmission of an earlier submission. The following is a list of the peer review reports and author responses from that submission.

Round 1

Reviewer 1 Report

In this study Ozer et al are reviewing the current clinical trials on Adoptive Cell Therapy for HCC. 

  The paper is very well written with steady narrative. The references are appropriate and up to date.    The introduction provides good insight into the subject and gives a good overview of the current situation and existing treatment modalities.   Overall, the paper provides a detailed and in depth review of the current trials (see tables) for CAR-T and TCR-T therapies.   No corrections warranted for publication. 

Reviewer 2 Report

This manuscript includes not enough data and discussion on adoptive cell therapy, even if are trials are ongoing.   There is a good introduction and abstract.  But the review itself is of insufficnet quality.

As it is now, the manuscript offers very little information on the topic.

You may consider answering the following questions:

- What is the scientific rationale that these new strategies may work?

- What are the challenges in terms of safety? There are already deaths described with this therapy.

- What about patient conditioning schedules prior to this therapy?

- You need to explain better the methodology to make it cristal clear for the reader.  The figure is not clear enough.

- A better critical analysis of current trials is needed, more than a list